# From Triboelectric Nanogenerator to Polymer-Based Biosensor: A Review

**DOI:** 10.3390/bios12050323

**Published:** 2022-05-11

**Authors:** Yin Lu, Yajun Mi, Tong Wu, Xia Cao, Ning Wang

**Affiliations:** 1Center for Green Innovation, School of Mathematics and Physics, University of Science and Technology Beijing, Beijing 100083, China; b20200369@xs.ustb.edu.cn (Y.L.); d202110423@xs.ustb.edu.cn (Y.M.); wut@nim.ac.cn (T.W.); 2National Institute of Metrology, Beijing 100029, China; 3Beijing Institute of Nanoenergy and Nanosystems, Chinese Academy of Sciences, Beijing 100083, China; 4School of Chemistry and Biological Engineering, University of Science and Technology Beijing, Beijing 100083, China

**Keywords:** polymer biosensors, triboelectric nanogenerator, self-power, energy harvesting

## Abstract

Nowadays, self-powered wearable biosensors that are based on triboelectric nanogenerators (TENGs) are playing an important role in the continuous efforts towards the miniaturization, energy saving, and intelligence of healthcare devices and Internets of Things (IoTs). In this review, we cover the remarkable developments in TENG−based biosensors developed from various polymer materials and their functionalities, with a focus on wearable and implantable self-powered sensors for health monitoring and therapeutic devices. The functions of TENGs as power sources for third-party biosensors are also discussed, and their applications in a number of related fields are concisely illustrated. Finally, we conclude the review with a discussion of the challenges and problems of leveraging TENG−based intelligent biosensors.

## 1. Introduction

With the fast development of social science and technology, people now require continuous personalized medical services [1,2,3,4,5,6]. On the one hand, biosensors have become essential for the early detection and management of many chronic and high-incidence diseases because of their high selectivity and sensitivity, fast analysis speed, chemical/mechanical stability, reusability, low limit of detection, and low cost [7,8,9,10,11]. On the other hand, long-term health monitoring and continuous health data analysis are highly appreciated in the clinical treatment of chronic diseases due to the sudden nature of diseases and accidents [12,13,14,15,16]. These days, great attention has been paid to the use of various types of wearable biosensing devices to obtain physiological data from the human body [17,18,19,20].

Wearable and implantable biosensing devices are gaining increasing interest for modern clinical treatment because they can simultaneously realize the multifunction of the observation, recording, and evaluation of various physiological activities [21,22,23,24]. Polymers promote the development of wearable and implantable biosensors due to their high flexibility [25], stretchability [10,26], self-repairability [27,28], biodegradation [29], and compatibility, which meet the requirements of wearable biosensors and implantable electronic devices in terms of both mechanics and biology [30,31,32].

In the era of IoTs, it is possible to deploy personalized biosensing devices on/in the human body to collect health data and treat human beings [33,34]. To achieve this, personalized wearable biosensing devices first need to integrate implantable sensing, therapeutical devices with power sources [35]. Furthermore, the health-monitoring system needs to be intelligent and miniature, which involves a variety of communication technologies and networking technologies to construct a large-scale and comprehensive “medical and health cloud” platform with adjustable accuracy [36]. However, current wearable medical sensing devices still face challenges in terms of balanced performance and sustainability [37,38]. For example, besides the concern of toxic chemicals and the large volume of rechargeable batteries, the short battery life is a source of constant complaint [39,40]. It is highly expected that biosensors can harvest and utilize thermal, solar, or mechanical energy from the environment and power themselves [41,42,43,44,45,46].

With the popularity of wearable devices, an energy harvester, triboelectric nanogenerator (TENG), is considered as a promising integrated element in line with the IoT trend [47,48]. Because of its unique materials selectable for fabrication and sustainable characteristics, it has been a hotspot since 2012 [49,50,51]. Moreover, based on a conjunction of triboelectrification and electrostatic induction, TENGs can convert various forms of mechanical energy in daily life into electrical energy [52,53,54,55]. To date, various research aspects for TENGs, such as mechanistic investigation, device development, performance optimization, in vivo applications, etc., have been widely studied [56,57,58]. It has been proven that various biomechanical movements can be used by TENG to generate electricity, including walking, arm flutters, breathing, heartbeat, pulse waves, sound vibration, blood flow, blood vessel pressure, gastric peristalsis, etc. [59,60,61,62,63,64,65]. At the same time, the generated triboelectric output variation caused by chemical or biological stimulation can be directly adopted as a sensing signal, thus introducing TENG−based self-powered sensor systems. For example, they can be applied in active sensing, therapeutic, drug delivery [66], limb movement monitoring [67,68], wound recovery, wireless transmission, and environmental monitoring [62,69]. This is a time for a systematic review with the aim to provide both an up-to-date summary of both recent advances and the challenges in the practical clinical application of TENG−based biosensing systems for personalized healthcare and IoTs.

Here, TENG−enabled polymer-based biosensors applied in healthcare and medical diagnosis systems are well-discussed by a cutting-edge depiction, as summarized in Figure 1. Biosensor devices are composed of different kinds of polymer materials, which have been used in physiological function regulation and health monitoring, including blood pressure, respiratory rate, nerve stimulation, etc. In addition, the usage of TENG as a power source for biosensors is also discussed, especially its wearable and implantable self-powered features, offering an insight into the variety of co-engineering alternatives and prospective capabilities of existing biological sensing devices. At the end of this review, challenges and perspectives for the development of intelligent biosensors integrated with TENG are also discussed.

## 2. TENG as Wearable Biosensors for Health Monitoring

In recent years, the TENG−enabled wearable biosensor (TWB) has come a long way [70]. The mandatory requirements of portable, comfortable, lightweight, flexible, high charge output, and even self-powering have rendered TWBs an extensive research opportunity [71,72,73,74]. With these properties, TWBs can be attached to different on/in-body positions to reflect various health conditions.

### 2.1. Body Condition Monitoring

Great advantages can be achieved by considering that TWBs can examine the state of human movement while non-destructively monitoring the metabolic level in real-time for multi-dimensional human health assessment. In this regard, sweat contains a lot of molecular biomarkers, including electrolytes, amino acids, metabolites, and hormones, and is an attractive medium for wearable sensors in analysis. Zhao et al. utilized ultra-stretchable Ecoflex fibers covered with multiwalled carbon nanotubes and polyaniline (PANI) as dielectric materials to fabricate TENG (Figure 2a) [75]. In addition, the fibers were modified with three types of enzymes so that the wearable device can sense the lactic acid, glucose, and creatinine concentrations in sweat in real-time to detect the metabolic level in vivo. When a person walks or runs, the movement of the body pushes the TENG to transmit an electrical signal, which may reflect the state of human movement. If perspiration wets a TENG, the electric signal containing the biomarker concentration information will be emitted because of the surface–triboelectric coupling effect. As shown in Figure 2b, the TENG can be woven into a textile as protective clothing, which brings a promising prospect in self-powered close-looped healthcare systems.

Qin et al. designed a fully flexible self-powered sweat sensor based on cellulose conductive hydrogel (Figure 2c) [76]. The hydrogel electrode was composed of a cellulose nanocomposite that included PANI, 2,2,6,6-tetramethylpiperidine-1-oxyl radical (TEMPO)-oxidized CNFs (TOCNF), and PVA/borax (PVAB). This TOCNF/PANI-PVAB hydrogel (CPPH) possesses superior stretchability (1530%) and self-healability (95% self-healing rate within 10 s without external stimulation), and high conductivity (0.6 S m^–1^) properties. Then, the CPPH electrode was encapsulated with PDMS and assembled with an ion-selective membrane (ISM) that can detect the Na^+^, K^+^, and Ca^2+^ concentrations. The TENG was operated in a periodic contact separation mode to generate signals, and the Na^+^-ISM and the PDMS layer encapsulating the flexible electrode acted as the positive and negative triboelectric layers, respectively. Figure 2d shows the real-time sweat-ion monitoring on a person during exercise on a mini stair stepper, and the Na^+^, K^+^, and Ca^2+^ current signals in perspiration during 30 min of exercise, which showed a steady increase in the signal magnitude when the person sweats. Furthermore, the quantitative analysis of the Na^+^, K^+^, and Ca^2+^ contents in perspiration showed high sensitivities of 0.039, 0.082, and 0.069 mmol^–1^, respectively.

Rao et al. developed a touch-sensitive electronic skin (e-skin) to simultaneously detect and discriminate between temperature and pressure from a TENG [77]. The e-skin’s bottom layer was a PDMS layer, and an electrode prepared by combining the Bi_4_Ti_3_O_12_ (BiTO) and rGO was prepared on the PDMS layer; then, silver paste was used as wires to lead out carbon nanotube fibers at both ends of the electrode film; finally, a PDMS triboelectric layer with a pyramidal micro-structured surface was the top layer (Figure 2e). The thermal-resistance electrode ensures that the e-skin can detect temperature and exhibit excellent properties (a high thermal sensitivity of β25/100:1024 K, temperature coefficient of resistance of 1.15% K^−1^ at 25 °C, and range of 25–100 °C). Furthermore, the tactile e-skin generated voltages in response to pressure and showed a superior pressure sensitivity of 5.07 mV Pa^−1^. Note that tactile e-skin can be worn on the human body to monitor the body condition (temperature and pressure) with the two signals not interfering with each other; as shown in Figure 2f, the tactile e-skin with 4 × 4 pixels was mounted on a person’s arm, and the recorded distributions of the temperature and pressure were presented. This work provides new pathways for portable detection and casts new light on the development of electronic skin.

### 2.2. Cardiology Sensors

Meng et al. first fabricated a woven-structured self-powered pressure sensor (WCSPS) for capturing subtle mechanical changes in the blood pressure in the vessel and expressing them as human pulse waveforms in the form of electrical signals [78]. For this TENG, polyethylene terephthalate (PET) was as one electrification layer, and indium–tin oxide (ITO) was added to the back of the PET to act as an electrode. Another electrification layer was a layer of polytetrafluoroethylene (PTFE) strips with an interlaced woven structure (Figure 3a). This pressure sensor achieved an ultra-high sensitivity (45.7 mV Pa^−1^ with an ultrafast response time of less than 5 ms) and there was no deterioration in performance after up to 40,000 motion cycles. Furthermore, two identical sensor systems (with integrated WCSPS and signal management circuitry) were respectively worn on the fingertips and the ears to detect human pulse waves, as shown in Figure 3b, and the detection of the pulse wave velocity (PWV) and blood pressure (BP) is shown in Figure 3c,d. In addition, a practical measurement was carried out on 100 people aged 24 to 82 who had different health conditions. The difference between the WCSPS-based blood pressure results and those provided by the commercial cuff device was approximately 0.87 to 3.65%. In the current complex cardiovascular monitoring systems, this work demonstrates a competitive, alternative way for low-power-consumption measurement. Working to enhance the durability of TENGs, Lou et al. designed a conductive polyurethane (PU) fabric-based TENG for pulse monitoring [79]. This sensing textile was made of a polyvinyl fluoride/silver (PVDF/Ag) nanowire nanofibrous membrane (NFM) and an ethyl cellulose NFM as the friction layers, and two layers of the conductive PU fabric as the electrodes (Figure 3e). The sensitivity achieved reached up to 1.67 and 0.20 V kPa^−1^ in the pressure range of 0–3 and 3–32 kPa, respectively. For practical use, this textile sensor can be fixed on the body parts you want, such as elbows, knees, and ankles, for dynamic motion detection. In addition, it can be positioned on the carotid artery to pick up impulse signals, providing a reliable way to reflect health status (Figure 3f). Moreover, this textile also collects data on pulse pressure by the amplitude of the P1 peak, and the reflected wave pressure generated from the hand can also be determined at the P2 peak.

Wang et al. developed a polyvinyl alcohol (PVA)-based triboelectric sensor to detect imperceptible skin deformation induced by human pulse and gather cardiovascular information [80]. PVA is among the most commonly used polymers for biomedical applications due to its water-solubility and bio-compatibility. Nevertheless, pure PVA has relatively limited functionality and high processing cost. In order to better understand the influence of the molecular/ionic engineering of PVA mixtures on their triboelectric performance, they utilized gelatin, KCl, or NaCl as the fillers to prepare PVA mixtures, and studied the impacts of factors such as pH, molecules, and ions. The results showed that the triboelectric devices constructed using the optimized PVA–gelatin composite films had stable and robust triboelectricity outputs. This PVA-based triboelectric device can be seamlessly worn on the human skin and produce pulse waveforms containing detailed pulse characteristics (Figure 3g). As shown, the heart rate can be determined as 76 beats per minute, and one of the enlarged pulse periods was for more in-depth analysis and health diagnoses. The basic understanding gained and demonstrated abilities allow for more capable biocompatible triboelectric devices that can continuously monitor vital physiological signals with high comfort.

As is well known, the proper choice of materials not only influences the manufacturing process, but also the final performance of the appliance. Dong et al. manufactured a skin-inspired stretchable and washable TENG (SI−TENG) for the surveillance of the arterial pulse [26]. In the SI−TENG system, a planar and designable conductive yarn network consisting of three strands of twisted silver-plated nylon yarn was embedded in a flexible elastomer. When the embedded yarn electrode was connected to human skin (i.e., the floor) through the external load, the SI−TENG would operate in a single-electrode mode and human skin could be regarded as another triboelectric material. As proven, real-time arterial pulse voltage signals can be achieved by sticking an SI−TENG (20 × 20 mm^2^) to a volunteer’s wrist (inset in Figure 3h), and the regular and repeatable pulse shapes were recorded under different physical conditions, such as normal and exercise. The heart rates of the tester were 67 bpm at rest and 88 bpm after exercise. The arterial pulse signal showed the three expected clear peaks, indicating that SI−TENG is able to monitor the pulse rate: percussion wave, tidal wave, and diastolic wave (Figure 3i). All in all, the self-powered SI−TENG could efficiently record pulse signals in real-time.

In another work by Xu et al., an ultra-sensitive pulse sensor on the base of TENG was developed for noninvasive multi-indicator cardiovascular monitoring [81]. In addition to the two triboelectric layers and electrodes, the self-powered ultra-sensitive pulse sensor (SUPS) in question also contained spacers and electrostatic shielding layers, and the whole device structures were finally wrapped by the encapsulating layers. Fluorinated ethylene propylene (FEP) film with a nanowire array and polyamide (PA) film with a fibrous structure were selected as the triboelectric layers. Cu foil served as the electrode material. Melamine sponge with a porous structure was used as the spacer. Additionally, the electrostatic shielding layer was made of Al foil attached to both sides of the SUPS. Between the Cu electrode and Al shielding layer was the FEP film assembled as an insulating layer. Finally, PET film was used to encapsulate the SUPS (Figure 3j). The SUPS held excellent sensing performances, including ultra-sensitivity of 10.29 nA kPa^−^^1^, low detection limit of 5 mg, and fast response time of 30 ms. When tested on a living subject, the monitored heart rates were consistent with the results of the commercial ECG. This device could be applied in the prevention and adjunctive therapy of cardiovascular diseases, such as arrhythmias, arterial rigidity, high blood pressure, and so on.

A more progressive approach to enhanced portability includes the integration of TENG in textiles. Ding et al. prepared a porous carbon nanotube (CNT)-doped PDMS film (300 mm × 300 mm) with a micropore structure as one of the friction layers [82]. The conductive textile was used as the electrode and aluminum as the triboelectric pair of PDMS. For biomedical applications, the TENG was applied to clothing and assembled into textiles, such as a sleeping belt (Figure 3k); then, it was placed below the human subject’s abdomen, and a heart rate of 70 bpm was recorded in real-time. This result shows that it can be used as a health monitor to diagnose illnesses and prevent emergencies.

Meng et al. reported a textile-based triboelectric sensor composed of a silver polyester fabric, an electrode, and a conductive hybrid flower-shaped textile for long-term portable biomonitoring [83]. It can achieve continuous and non-invasive measurements of human pulses. With a sensitivity of 3.88 VkPa^−^^1^, it can measure pulse waves in elderly and weak people. Based on the continuous monitoring of the pulse wave all night and the illness of the breath, it can be accurately identified in time. Similarly, Chen et al. proposed a noncontact heartbeat and breathing detection system based on a self-powered pressure sensor (SPS) with flexible hollow microstructures (HMs) [84]. For the structure, a folded FEP/Ag sheet was the outer sheath and inner HMs, and an ethylene-vinyl acetate (EVA)/Ag sheet was the core. By placing the sensor strip underneath the back of the lying tester without skin contact, the sensor can be driven by repeated pressure variation due to the slight thoracic movement during inhalation and exhalation and deliver a time-varying voltage output. A pressure sensitivity of 18.98 V kPa^−1^ and a wide working range of 40 kPa can be achieved. It is worth mentioning that the as-prepared sensor not only has similar results to the ECG, but can also operate without skin contact, which demonstrates promising feasibility in noninvasive clinical and medical applications.

Fan et al. introduced a TENG−based all-textile sensor array (TATSA) with high sensitivity for epidermal subtle pressure capturing [4]. Figure 3l shows that the TATSA was sewn into a piece of cloth, which was knitted using the conductive yarn and commercial nylon yarn together in a full cardigan stitch. This sensor exhibited excellent properties, including a high sensitivity of 7.84 mV Pa^−1^, fast response time of 20 ms, high stability of over 100,000 cycles, and wide working frequency bandwidth. Notably, this TATSA can be directly integrated into different sites of the fabric, such as at the neck, fingertip, wrist, and ankle positions, to obtain the corresponding pulse signals as well as the respiratory wave signals in the abdomen and chest.

In particular, for textile-based sensors, they still have a limitation in preparing more delicate structures on a large scale because of the mechanically weak conductor and polymer coatings on existing fabrics and threads. Shin et al. explored a portable PVDF fiber-based sensor for cardiovascular detection using a sewing machine method to produce programmable textile motifs [85]. A dry-jet wet-spun PVDF fiber was used to provide the mechanical strength to allow the sewing machine to stitch them into various programmable textile patterns on different fabric substrates, promoting its aesthetic appeal for different designs. The prepared multi-ply PVDF fibers showed a tensile strength of 0.24 GPa and tensile modulus of 4.08 GPa; the performance was higher than that of conventional PVDF fibers. The sensor stitched into 3 × 10 mm^2^ could provide a sensing range of 326 Pa–326 kPa, with sensitivities of 6.23 mV kPa^−1^ and 1.12 mV kPa^−1^ at pressures below and above 16.3 kPa, respectively. For cardiovascular sensing, the sensor was applied to the neck to monitor arterial pulse, and successfully extracted systolic and diastolic pressures (Figure 3m). Moreover, the device can also monitor the pulse rate in real-time, and a distinguishable increase in current after exercise indicates increased cardiac output, as seen in Figure 3n, o. This work demonstrates a workable manufacturing approach for making textile sensors sewn using a sewing machine, with many potential e-textile applications.

### 2.3. Pneumology Sensors

The ability to monitor respiratory conditions is a very valuable attribute, especially during fast-spreading novel coronavirus disease [86]. Lu et al. designed a TENG assembled into a facemask for respiratory sensing. When a human wears the smart facemask and breathes, the TENG on the facemask can generate a maximal output voltage of 8 V and a maximal output current of 0.8 μA. The detailed structure is shown in Figure 4a. Driven by the air flow, the FEP film and Al foil will make contact when the person inhales. Moreover, the smart facemask was integrated into a smart human–machine interface (HMI) system that was also driven by breath; in this way, people can control small household appliances conveniently. In addition, an apnea alarm system was further constructed to provide a real-time alarm after breathing stops. This work could significantly lower the financial and human costs of respiratory surveillance and stimulate the development of the medical field.

He et al. designed a TENG sensor for monitoring respiration using two nanofiber layers, polyacrylonitrile (PAN) and PVDF, which can be attached to a commercial mask to monitor the respiratory status. This sensor can monitor multiple respiratory indices, such as the respiratory rate (RR), inhalation time (tin), exhalation time (tex), and their ratio (IER = tin/tex) with high filtration efficiency [87]. The monitored RR and IER showed 100% and 93.53%, consistent with the real-time RR and IER set on the ventilator, respectively. Furthermore, it had a filtration efficiency of 99 wt% for particle sizes between 0.3 µm and 5 µm. This work has promising potential for self-powered health diagnostics.

In 2019, Qiu et al. reported a more comfortable wearable device that could be used as a respiratory monitor and a real-time communicator [88]. The designed TENG was based on ordinary fabrics. The PANI was firstly polymerized in conventional cloth and acted as an electrode, and then polycaprolactone (PCL), thermoplastic polyurethane (TPU), and PVDF were covered in the proper sequence by electrospinning to prepare the TENG negative material. Similarly, the positive layer of TENG was prepared on a conductive cloth, followed by electrospinning PCL, TPU, and PA6 in the proper order. With this TENG, a 47 μF commercial capacitor can be charged to 4.5 V. Furthermore, this TENG can be used for human respiratory monitoring, as shown in Figure 4b–d. Normal breathing, deep breathing, and rapid breathing could be distinguished clearly; furthermore, an alarm would be triggered when the tester’s respiratory signal was below the threshold. This work gives a more natural way for wearable health monitoring without connecting any external equipment, but with certain portability and comfort.

Another respiratory motion sensor proposed by Han et al. was a fish gelatin (FG)−based TENG that could be attached to clothing for measurement [89]. The TENG was composed of FG film and a PTFE/PDMS composite film, and showed improved performance: the open-circuit voltage, short-circuit current, and output power density of this FG−TENG with a size of 5 × 5 cm^2^ could reach up to 130 V, 0.35 μA, and 45 μW cm^–2^, respectively. Importantly, the results showed that FG−TENG could monitor shallow and deep breathing when integrated into clothing. Owing to its superior design, the FG−TENG had the capability to sense physiological conditions, even those particularly small in size.

Obstructive sleep apnea and hypopnea syndrome (OSAHS) is a respiratory illness caused by the obstruction of the upper respiratory tract, which affects the quality of sleep and human health. Real-time respiratory surveillance can help detect apnea symptoms and provide early warning, diagnosis, and appropriate treatment. Zhang et al. proposed a self-powered, portable, TENG−based, waist-sized respiratory surveillance device [90]. The respiration monitoring system was comprised of a belt with a TENG−based respiration sensor built within it and a wireless transmission unit. The TENG sensor retrieves respiratory information by detecting changes in the abdominal circumference during breathing, as shown in Figure 4e–g. In this TENG, commercial nylon and PTFE films were used as dielectric materials, and two copper sheets were attached to the layers of dielectric material as conductor electrodes. A series of breathing tests demonstrated the correlation between the electrical output signals and respiratory processes in the chest and abdominal mode of breathing. Recently, Peng et al. designed another self-powered electronic skin (SANES) based on a TENG for OSAHS diagnosis [91]. PAN and “polyamide 66” (PA 66) nanofibers were selected as the contact pairs, and the deposited gold was used as an electrode. The e-skin showed a peak power density of 330 mW m^−2^ and high pressure sensitivity of 0.217 kPa^−1^. The SANES was attached to a personal abdomen, and the undulating breathing motion could be detected by electrical signals in real-time. The e-skin was also used during sleeping to detect apnea and hypopnea and achieve assessment. These works pave a new and practical pathway for personal respiratory health monitoring and sleep breathing disease clinical detection.

Lin et al. introduced a large-scale, pressure-sensitive, and washable intelligent textile based on a pressure-sensitive TENG network as a bed sheet for monitoring sleep behavior in real-time and clinically derived alarm [92]. As shown in Figure 4h, out of the smart textile were three layers: the top and bottom layers were sheets of conductive fibers and the middle layer was a wavy PET film. The textile exhibited an excellent pressure sensitivity of 0.77 V Pa^−1^, fast response time of less than 80 ms, and high mechanical durability. To obtain more detailed information, a unique software algorithm was used next to a network of 4 × 4 sensors to track the movement and strength on the top of the sheet. This work not only opens up a new pathway for non-invasive real-time sleep monitoring, but also provides a new perspective for practical applications of remote clinical medical services.

Zhou et al. introduced an ultra-soft, single-layer intelligent textile composed of washable functional fibers for continuous sleep surveillance and early intervention in sleep-related illnesses [93]. The lead wire was formed by twisting polyester threads around a 10 μm-diameter stainless steel rod. As Figure 4i–l shows, the intelligent fabric consisted of 61 detection units and exhibited a high-pressure sensitivity (10.79 mV Pa^−1^), good stability, and a wide working frequency bandwidth of 0 to 40 Hz. The device was also tested in healthy individuals compared to OSAHS; for the healthy person, the flow of air into and out of the lungs was stable, and showed a normal breathing rate. However, for OSAHS patients, when their soft tissue began to block air from passing through the trachea and cause instigating OSAHS events, the device recorded abnormal respiratory rates and a clear apnea of 11 s that can be seen in Figure 4m–r. It should be widely used in domestic and clinical health systems in the near future and could change traditional methods of sleep surveillance.

**Figure 4 biosensors-12-00323-f004:**
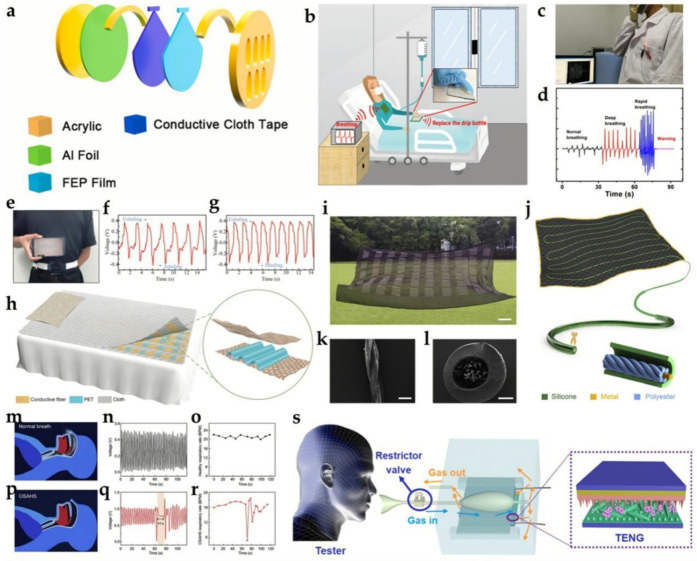
Applications of wearable TENG−based biosensors in pneumology. (**a**) Detailed structure of respiratory-sensing TENG. Reprinted/adapted with permission from Ref. [86]. 2022, Lu et al. Elsevier. (**b**) Wearable TENG for intensive care. (**c**,**d**) Photograph of respiratory monitoring and the corresponding output current. Reprinted/adapted with permission from Ref. [88]. 2019, Qiu et al. Elsevier. (**e**) Photograph of volunteer’s respiratory waveform displayed on a mobile phone via the wireless transmission system. (**f**,**g**) Displayed monitoring signals respectively from the 22-year-old volunteer and the 24-year-old volunteer. Reprinted/adapted with permission from Ref. [90]. 2019, Zhang et al. Elsevier. (**h**) Illustration of a 9 × 11 TENG−array−based smart textile. The inset is an enlarged view of one TENG unit of the smart textile. Reprinted/adapted with permission from Ref. [92]. 2018, Lin et al. WILEY-VCH. (**i**) Photograph of the smart textile with 61 sensing units. The scale bar is 10 cm. (**j**) Schematic illustration of a sensing unit, which was fabricated by weaving a washable functional fiber into a black textile substrate with serpentine structure. (**k**) SEM image of the conductive fiber (Scale bar: 300 μm). (**l**) Cross-section view of the washable functional fiber (Scale bar: 250 μm). (**m**) Schematic illustration of the smooth airflow into and out of the lungs for a healthy participant, and the corresponding (**n**) respiratory signal and (**o**) respiratory rate. (**p**) Illustration depicting falling soft tissue blocking air from passing through the trachea, inducing OSAHS. (**q**) Respiratory signal and (**r**) respiratory rate of an OSAHS participant. Reprinted/adapted with permission from Ref. [93]. 2020, Zhou et al. Elsevier. (**s**) Schematic illustration of a human respiration-driven system. Reprinted/adapted with permission from Ref. [94]. 2019, Wang et al. Elsevier.

Wang et al. developed an NH_3_-sensitive sensor based on a respiratory TENG, proving the feasibility of TENG used in the preliminary diagnosis of human renal health (Figure 4s) [94]. A Ce-doped ZnO–PANI nanocomposite film was selected as the triboelectric layer as well as the sensitive layer, and a PDMS film was used as another triboelectric layer. This TENG exhibited excellent ability for respiratory flow monitoring (2–6 L min^−^^1^), and the output voltage NH_3_ sensor monotonously decreased with the increase in the NH_3_ concentrations from 0.1 to 25 ppm. It is interesting that the TENG showed a good response (21%) when the tester had buccal ulcers.

Xue et al. developed a new flexible smelling e-skin from PANI/PTFE/PANI sandwich nanostructures based on the triboelectrification/gas-sensing coupling effect to detect ethanol [95]. The flexible e-skin smell can be easily controlled by human movement/breathing, and the output/voltage current was greatly dependent on the environmental atmosphere. The gas flowing from human heavy breathing or blowing vibrates the stretchable middle PTFE membrane to repeatedly contact and separate with top and bottom PANI films, which efficiently drove electrons to flow back and forth because of the unbalanced electric field difference. For ethanol, the electronic skin detection limit was 30 ppm, and the response was as high as 66.8 versus 210 ppm for ethanol gas. In addition, this smelling e-skin can be used for visually identifying drunken drivers. The detecting system consisted of a gas-flow processor, a smelling e-skin, and a visualization panel, where the LED bulbs on the visualization panel could be used to clearly indicate the drunkenness level of the tester by just simply counting the number of lit LED bulbs. Their results may lead to a new direction of research for the development of e-skins with specialized functions.

### 2.4. Motion Sensors

#### 2.4.1. Foot and Leg Motions

Foot strikes are both typical health indicators and a rich energy source, with a potential power of 67 W for walkers [96]. Various TENGs have been actively studied for footstep energy conversion. Since the steps typically experience a force directed towards the ground, TENG using the contact separation mode are widely used for this conversion of motion energy. In particular, multi-layer structures have attracted considerable attention because of their effectiveness in improving production performance [97]. In 2016, Zhu et al. reported a 3D knitted spacer fabric carefully engineered from TENG using vertical contact electrification between two polymers with different friction polarities. The open-circuit voltage for a TENG pixel was over 3 V, the short-circuit current was approximately 0.3 μA, and the output power was 16 μW. The TENG−based 3D spacer fabric set can easily power LEDs. More importantly, it can also be used as an automatic monitoring system to track and recognize human motion. Furthermore, it can also sense the pressure of feet in situ with high sensitivity during human walking. This method shows excellent output performance and excellent intelligent sensing ability [98]. Recently, the Jasim Uddin research group presented a TENG based entirely on nanofibers for energy gathering and biomechanical detection applications [25]. TENGs were prepared using a forced spinning (FS) method to produce PVDF and thermoplastic polyurethane (TPU) nanofiber (NF) membranes. The TPU nanofiber film was attached to a uniformly sputtered gold nanofilm. The experimental characterization of PVDF–TPU/Au–NF–TENG showed that, under the load of 240 bmp, the maximum open-circuit voltage and short-circuit current generated by the surface in contact with the dispersed gold in the TPU fiber membrane were 254 V and 86 μA, which were higher than those of bare PVDF–TPU and NF-based TENG by 112% and 87%, respectively. This TENG can light up 75 LEDs (1.5 V each) by clapping hands. The PTA–TENG also had sensing properties for different body movements, such as bicep contraction signals for pound lifts and incline push-ups, leg up and down movements, and thigh muscle (quadriceps) contraction signals for jogging and running.

Lin et al. proposed a TENG based on a contact-separated mode seesaw structure (SS−TENG) [99]. The TENG mainly consisted of a frame with four springs, a top triboelectric unit (TTU) attached to the frame, and a bottom triboelectric unit (BTU), which were connected to two seesaw structures through flexible connectors. Due to this design, when the TTU was dragged downwards by an external force, the seesaw (SSL)-structured links could be triggered and the BTU lifted up. With this design, the two friction surfaces will move in contact or separation at the same time. Compared with the previously reported contact–separation-mode TENG, the relative motion speed between the two friction surfaces will be greatly increased, contributing to a substantial increase in the electrical output. The sequence and time interval of the asymmetrical output signal may reflect the direction and speed of the object in motion. If the equipment were installed in standard footwear, the movement of the human foot could be monitored (Figure 5a–c).

Currently, problems related to the efficient production of energy on foot by TENG primarily include: (1) wearing comfort, (2) high power, and (3) mechanical durability. This competitive power generation capability holds great promise for continuously driving biomedical devices for personalized healthcare, such as pulsed wave sensors, not counting on external batteries in the future.

The comfort and versatility of the insole are equally important. For example, compressibility allows users to walk without any discomfort. Yang et al. designed an in-shoe sensor pad (ISSP) connected to the upper, which was based on a TENG, to monitor the real-time stress distribution on the top of the foot [97]. Each ISSP sensor unit was a TENG (AC−TENG) air cap consisting of an activated carbon/polyurethane (AC/PU) and microsphere array electrode (Figure 5d,e). The detection range for each AC−TENG was 7.27 MPa, which was sufficient for monitoring the change in pressure during different movements. The multifunctional ISSP can realize many typical functions of traditional intelligent shoes, including step calculation, human–computer interaction, and so on. In addition, it can reveal special information, including the shape of the shoes, the concentration of pressure on the toes, and comfort in sport. The signal-processing and data-transmission module in the system adopts the hybrid power supply of a wireless power supply, which can observe the foot information in real-time on a mobile phone. Because the insole was constantly in contact with the rough ground during walking, the problem of mechanical stability may occur. Developing TENG with high wear resistance and high flexibility and toughness is a good solution.

Sun et al. reported a simple way to build a TENG−based self-powered sensor to monitor human movements in the fire field. Patterned MXenes ink electrodes (MIEs) were written directly on triboelectric materials [103]. A commercial aramid non-woven fabric and heat-resistant silica gel were used as the positive and negative materials of triboelectricity, respectively. The MXenes ink electrode was chosen to write directly on the back of the aramide fabric, and a patterned circuit was built to collect more information about human movements. The equipment had good high-temperature resistance. The multi-arch TENG was attached to the arm to monitor the arm movement. With the increase in bending angle (0°–140°), G−TENG showed good sensitivity (0.11 V degree^−^^1^) and could recognize simple emergency gestures (stop, turn right, and turn left). Secondly, more importantly, the G−TENG sensor provided good performance in sensing pressure distribution and different gaits. With the help of machine learning (ML) processing signs and fire shoes equipped with G−TENG sensors, seven gaits (walking on a flat road, going downhill, running, jumping on the right foot, jumping on both feet, up the stairs, and down the stairs) were successfully perceived and recognized.

#### 2.4.2. Arm Motion

Sheng et al. reported a double-network polymer ionic conductor sodium alginate/zinc sulfate/polyacrylic acid–acrylamide (SA-Zn) hydrogel TENG (SH−TENG), which exhibited excellent stretchability (>10,000%), high transparency (>95%), and good electrical conductivity (0.34 S m^−1^) [5]. The SA-Zn hydrogel-based TENG had excellent electrical properties, and 234 commercial green LEDs can be instantly lit by touching and pressing the SH−TENG with a contact area of 4 × 4 cm. The optimal performance of SH−TENG exhibited a good linear response with tensile force, so they developed a conductive hydrogel-based SH−TENG self-powered elastic training band sensor to monitor the data during stretching motion. The proposed SH−TENG may have potential applications in medical surveillance and rehabilitation, electronic skin, autonomous sensors, and man-computer interaction in the future.

Wang et al. designed a stretchable, shape-adaptive liquid single-electrode triboelectric nanogenerator (LS−TENG) based on potassium iodide and glycerol (KI-Gly) liquid electrolyte as the working electrode, which was used to collect human motion energy and power wearable electronic devices [104]. LS−TENG exhibited high output performance (300 V open-circuit voltage, 17.5 mA m^−2^ short-circuit current density, and 2.0 W m^−2^ maximum output power) and maintained steady output performance. It would not deteriorate under 250% tensile tension and after 10,000 repeated contact separation movements. In addition, LS−TENG could collect biomechanical energy, including rocker arms, human walking, and knocking hands, to power commercial electronic devices without an additional power supply. LS−TENG could also be attached to different joints of the body and be used as a self-powered human motion monitor.

Saadatnia et al. proposed a high-performance TENG based on nano-scale porous polyurethane aerosols (PUAs) for efficient mechanical energy collection and biomechanical detection [100]. The electrical output properties of PUA−TENG were investigated using aerogel films with different porosity levels, ranging from 0 to 94% open pore content. PUA−TENG with 33% open-pore content exhibited the highest performance, with a 3.5 times higher open-circuit voltage and short-circuit current than TENG with a non-porous membrane. The PUA−TENG was capable of charging various capacitors under mechanical agitation for power collection applications (Figure 5f). The device can be mounted on a human arm to monitor arm motion as a biomechanic sensor.

#### 2.4.3. Hand Movement

Zhao et al. developed a flexible single-electrode TENG (S−TENG) based on polyester conductive cloth as a working electrode to harvest human motion energy to power LEDs and portable electronic devices [105] and flat conductive fabric enveloped in a soft elastomer. Skimmed cowhide was first selected as the tribo-positively charged material for TENG. When the fabricated S−TENG measured 40 × 100 mm^2^, high output performance was achieved, producing an open-circuit voltage of 534 V and a power density of 230 mW m^−2^ at an operating frequency of 3.0 Hz. With an integrated rectifier, the S−TENG could power 240 LEDs, charge various capacitors, and drive electronic watches or calculators. Furthermore, S−TENG can harvest biomechanical energy from wrist movements, hand taps, and human walking. Meanwhile, S−TENG, as a self-powered sensor, could be used to monitor subtle signals of human physiological activities, such as finger movements, facial masseter muscle activity, and diaphragmatic breathing. In addition, S−TENG could be tied to clothing (such as wool coats or nylon sweaters) to gather energy from wrist movement.

Gloves are another attractive candidate due to their flexibility/wear resistance, easy use, low cost, streamlined signal processing, and low energy consumption [106]. For example, He et al. reported an intuitive, autonomous, cheap, and glove-based HMI [101]. A minimalist design based on two configurations of TENG was proposed to balance the requirements for all functions of HMI and simplified signal-processing capability. Each TENG sensor consisted of a PEDOT:PSS-coated textile belt sewn to the glove and a layer of silicone rubber film coated on the glove. The glove-based interface has successfully demonstrated wireless car control, wireless drone control, minigame control, VR game control, cursor control for online shopping, and letter writing through simple and intuitive operation skills.

Various types of patches and canes are also used to generate electricity for the fingers. Xu et al. designed an antifreeze organo-hydrogel absorbed by a polyacrylamide/nano clay network and ethylene glycol (EG)/water [107]. The antifreeze binary solution provided the organo-hydrogel with excellent properties at 30 °C, including a tensile modulus of 29.2 kPa, an ultimate tensile strain of 700%, an ionic conductivity of 1.5 × 10^−3^ S m^−1^, 91% transparency, and fast self-healing. Flexible organic hydrogel electrodes were assembled with elastomers to prepare TENG and then further attached to fingers to develop human–machine interaction keyboards. The voltage signals generated by the keyboard in contact with multiple surfaces were collected, coded, and interpreted as letters and punctuation marks, which were then displayed on the monitor. It also showed typing with a self-powered flexible keyboard at −30 °C.

In 2018, Cao et al. reported a triboelectric sensor (SNT) based on nanofibers for respiratory health surveillance. Screen-printed silver nanoparticle (AgNP) electrodes and electro-spun PVDF nanofilms constitute SNTs with an arched structure [108]. The air permeability of A4 paper was 1.4 mm s^−^^1^, while the air permeability of this SNT was as high as 6.16 mm s^−^^1^, which was much higher than that of the traditional cast film. When folded, bent, or even twisted, it also showed extraordinary softness so that the characteristics of the skin ensured its excellent electronic performance. In addition, because of their excellent performance, SNTs have been considered as finger motion trackers and for predicting activity.

Recently, Li et al. prepared a new customized polyvinyl alcohol nanocomposite polymer (NPE) polymer PCD/electrolyte modified polycation (PCD) point and used it as the primary triboelectric material to construct a new NPE−TENG [109]. First, a new NPE was prepared by adding modified polycation CDs (PCDs) to the PVA matrix; then, the NPE was coated with conductive silver-plated nylon yarn (NPE@SPNYs). The introduction of PCDs can significantly improve the triboelectric properties of NPE fibers by improving the ionic conductivity. Based on NPE fibers, a core-sheath-fiber TENG was further developed for efficient biomechanical energy collection and multifunctional all-round personal health monitoring. The developed NPE−TENG could respond to different mechanical stimuli with excellent flexibility and provide a high power density of 265.8 µW m^−1^. A wearable sensor based on self-powered NPE−TENG was further designed, which could realize a fast, real-time, and non-invasive way from physiological signals (subtle expression, breathing, and vocal cord vibration) to joint movement (wrist, elbow, and knee). In addition, a self-powered smart glove was designed for gesture detection and recognition. As a power supply, NPE−TENG can charge commercial capacitors, operate small electronics, and turn on hundreds of LEDs.

#### 2.4.4. Muscle Stretching

The periodic muscle contraction/relaxation process can also be used directly on the skin to generate electricity using TENG. Deng et al. proved a similar idea and developed a carbon nanotube (CNT)-silicone rubber liquid composite with excellent conductivity and fluidity, which could achieve 900% tensile deformation, and successfully realized the integration of biomechanical energy collection and multifunctional sensing (Figure 5j) [102]. For active sensing, the dynamic motion of human joints was detected due to the correlation between gestures and corresponding electrical signals (Figure 5k,l). Finally, a self-powered wearable keyboard based on an SS−TENG array was shown, which had excellent consistency on the surface. Therefore, for these movements with low power-generation capacity in terms of output performance, the converted electrical signal itself can be used as sensing data for biological activity monitoring through self-power supply.

## 3. TENG as Implantable Biosensors

Implantable medical devices play an important role in daily disease monitoring and treatment. However, these medical devices have limited battery life [110,111]. Replacing the battery surgically is not only an expensive and risky procedure for the patient, but also causes pain and inconvenience to the patient. The battery itself contains toxic chemicals and is not an ideal power supply for implantable medical devices. Given this, self-powered implantable medical devices are a promising, long-term solution and are being studied in increasing detail. TENG device materials come from a wide range of sources, are biocompatible, can be miniaturized and weight-reduced according to the needs of use, and can also be encapsulated to prevent rejection and infection. These excellent properties lay the foundation for the development of new implantable power sources. Additionally, the passive energy of the human body in daily life is inexhaustible. This section demonstrates the feasibility of implantable biomedical sensors based on TENG through cardiology, pneumology, and other fields.

### 3.1. Cardiology Sensors

Implantable biomedical sensors are used to monitor cardiac conditions in patients with a history of heart disease and to diagnose patients with cardiovascular disease [24,112,113]. Encapsulated by biocompatible or biodegradable materials, TENG has the advantages of high sensitivity and low cost, and is commonly used as a self-powered biosensor for internal pressure heart rate monitoring or as a power source for stimulation applications, such as pacemakers.

Inspired by biological symbiosis, Ouyang et al. demonstrated a fully implantable symbiotic pacemaker (SPM) based on an implantable triboelectric nano-generator (I−TENG), which can perform energy collection and storage, and large-scale animal-scale cardiac stimulation [114]. The symbiotic pacemaker has been successful in correcting and preventing sinus arrhythmia. Using the contact separation mode, the core-shell structure of I−TENG was composed of two triboelectric layers, a support structure, and a shell with two packaging layers (Figure 6a,b). Nanostructured PTFE films were used as negative triboelectric layers. Three-dimensional (3D) elastic sponge (ethylene vinyl acetate copolymer, EVA) acted as a spacer, and a memory alloy belt (high elastic titanium) was used as a keel. I−TENG was completely encapsulated by a flexible Teflon film and PDMS layer to enhance its structural stability and avoid damage to the equipment caused by environmental liquid. The average values of V_oc_ (open-circuit voltage), Q_sc_ (charge transfer), and I_sc_ (short-circuit current) of I−TENG were 97.5 V, 49.1 nC, and 10.1 μA, respectively. When implanted into the organism, the V_oc_ in the body was up to ~65.2 V, the Q_sc_ was ~13.6 nC, and the corresponding I_sc_ was ~0.5 μA. The energy collected from each cardiac cycle was 0.495 μJ, which is higher than pigs’ (0.262 μJ) and humans’ (0.377 μJ) pacing threshold energies. In addition, I−TENG showed excellent mechanical durability (100 million mechanical stimulation cycles) and cell compatibility, which were the determinants of the long-term implantation of devices. However, I−TENG and SPM still need to overcome some challenges to achieve clinical application. For example, in order to meet the requirements of a minimally invasive implantation process and improve the comfort of long-term in vivo surgery, I−TENG with the characteristics of small volume, high energy density, effective fixation with biological tissue, and long-term biosafety needs to be developed.

Liu et al. reported a small self-powered TENG−based endocardial pressure sensor (EPS), which was highly sensitive and could be used to detect arrhythmias [115]. TENG presented a multilayer structure, in which the spacer was sandwiched by PTFE film treated by inductively coupled plasma, and the gold electrode was deposited on the back and aluminum foil, and then encapsulated with a PDMS layer. Using a retractable heparin-coated polyvinylchloride catheter, minimally invasive implantation of SEPS was carried out in adult male pigs in Yorkshire (Figure 6c,d). It was used to record ECG, the femoral artery pressure (FAP), and working signals, with a sensitivity of 1.195 mV mmHg^−1^. The device has good mechanical and electrical stability at the same time, and can also detect arrhythmias, such as ventricular fibrillation and ventricular premature beats. These advancements make it possible to safely perceive stress, diagnose and monitor cardiovascular disease, and show broad prospects in the field of implantable medical surveillance.

Similarly, Ryu et al. explored a high-performance inertia-driven triboelectric nanogenerator (I−TENG) based on body motion and gravity [120]. Based on body motion and gravity, the generator used amine-functionalized poly-(vinyl alcohol) (PVA-NH_2_) and perfluoroalkoxy (PFA) as triboelectric materials to obtain a cylindrical I−TENG with a radius of 1.5 cm and a height of 2.4 mm. The author also successfully used I−TENG stacked in vivo in the preclinical test and gathered real-time output voltage data through the low-power Bluetooth information-transmission system. In addition, a self-charging pacemaker system using I−TENG to charge its battery was successfully demonstrated. The synchronous stacking structure to realize the superposition of current waveform can gradually increase the peak current value without additional components so that the I−TENG stacked in five layers can produce 136 V_peak_ and 2 μA_peak_ cm^−3^. Future manufacturing work may incorporate TENG’s better power performance into self-powered body-implantable medical devices and health monitoring systems to improve the welfare of patients.

Another work on implantable wireless self-powered heart detection was completed by Wang Z. L.’s team [116]. Nanostructured polytetrafluoroethylene (n-PTFE, 50 μm) was used as a triboelectric layer to enhance the output signal. Kapton film was used as a flexible substrate to allow contact separation in I−TENG. An ultra-thin gold layer (50 nm) was deposited on the back of the Kapton film to form an electrode. Aluminum foil (100 μm) acted as another triboelectric layer and another electrode. The device showed a multi-layer structure and had excellent in vivo performance and stability. Driven by a pig’s heartbeat, its V_oc_ was up to 14 V and I_sc_ was 5 μA. I−TENG continuously generates power within 72 h of implantation, showing that it has excellent durability, and successfully transmits electrical signals related to heartbeats in vivo by using a self-powered wireless transmission system (SWTS) (Figure 6e–h). In view of its high output performance and long-term reliability, the device was considered to be significant progress in the power supply of implantable medical devices.

Ma et al. demonstrated a self-powered multifunctional implantable triboelectric active sensor (ITEAS) capable of permanently monitoring a variety of physiological and pathological signs [121]. ITEAS was inserted into the pericardium of live pigs and attached to the pericardium. It produced a V_oc_ of ~10 V and I_sc_ of ~4 μA. At the same time, the core-shell packaging strategy allowed it to still exhibit excellent monitoring function and high biocompatibility within 72 h after closing the chest. This demonstrates that it has great potential for the future of the healthcare industry.

In 2021, Shao et al. developed a new environmentally friendly in-situ gap-generation method to manufacture a non-isolated triboelectric nanogenerator (NSTENG) [122]. NSTENG consisted of a layer of copper as the electrode, a rubber layer on the outside, and a gap generated by evaporating distilled water soaked in copper. When mounted on the rat heart, the movement of the heart lead to periodic contact and separation between the two friction layers of NSTENG, producing an electrical output containing rich information regarding cardiac activity. The anterior wall (AW) and left wall (LLW) of the heart were implanted to detect cardiac motion. Voltages of 0.11 V and 0.15 V were generated in the AW and LLW of the heart, respectively. NSTENG’s heart rate accuracy measured at LLW was up to 99.65%, and it could detect the subtle movement of the heart that cannot be captured by ECG, indicating its important role in providing important supplementary information for the diagnosis, treatment, and prevention of cardiovascular diseases other than traditional ECG.

TENG−based implantable heart sensors provide an autonomous power strategy that uses the body as a sustainable energy source. Additional efforts are needed to optimize output performance, power transfer, long-term stability, and energy storage density. Continued miniaturization and improvements in sensor technology will help develop new equipment and open the door to more innovative and less costly cardiovascular surveillance methods.

### 3.2. Pneumology Sensors

In most current TENG−related solutions, respiratory monitoring is only a secondary focus, and the main goal is to use in vivo body movement as an external energy source; some works have been conducted on the in-depth research on implantable sensors. Therefore, the review in this section will cover the available technologies in this field.

In 2014, Zheng et al. demonstrated in vivo biomechanical energy collection by first using TENG [123]. I−TENG has been developed in live mice to collect the energy of their periodic respiration. The energy generated by breathing was directly used to power a prototype pacemaker. The I−TENG consisted of PDMS, deposited Au (50 nm), and thin and Kapton substrates (30 μm). The I−TENG was implanted subcutaneously into the left chest of rats. Inhalation and exhalation by rats led to alternating expansion and contraction of the chest, which in turn produced the deformation of the thin Kapton film, resulting in periodic contact and separation between the PDMS nanostructures and Al foil, respectively. The amplitude of the breathing-generated voltage and current signals were approximately 3.73 V and 0.14 µA, respectively, and the power density was up to 8.44 mW m^−2^. This work has also taken a key step for the development of implantable self-powered medical equipment.

In 2018, Li et al. developed an in vivo micropower supply based on a breath-driven implantable TENG [117]. It can convert the slow diaphragm machine movement during breathing into a high-frequency AC output. The device consisted of a multilayer structure, including top and bottom micro-interdigitated electrodes, an intermediate layer, and a soft silicone elastomer package with a cavity design. The entire device had an ultralow Young module of approximately 45 kPa and high biocompatibility to soft biological tissues. When implanted in the abdominal cavity of Sprague Dawley rats, diaphragm motion during normal breathing can be converted into a continuous ≈2.2 V direct current (DC) output (Figure 6i–k). With the help of breath-driven TENG, a light-emitting diode can be continuously powered without decay. This improved implantable TENG structure may provide a promising solution for the development of self-powered IMDs.

At present, the research on TENG equipment is becoming more and more in-depth, but the analysis of pneumology-related medical equipment is not as concentrated as that in other fields. Although most studies have studied respiratory surveillance in parallel, their main research continues to focus on obtaining passive respiratory biomechanics as a source of energy. In view of the scarcity of published papers in this field, extensive research and investment are still needed in this field to promote the development of the next generation of small implantable sensors for the diagnosis of lung diseases.

### 3.3. Other Implantable Biomedical Sensors

In order to inhibit orthopedic implant-related infection and promote bone formation on the implant’s surface, Tian et al. proposed a self-powered flexible implantable electrical stimulator, which has greatly enhanced osteoblast attachment, proliferation, and differentiation [19]. The self-powered stimulator was composed of TENG, a rectifier, and a flexible interdigital electrode. PTFE and indium tin oxide (ITO) were selected as the friction layer, and then the integrated TENG and interdigital electrode were passed through a thin flexible PDMS layer (50 μm) encapsulation. As shown in Figure 6l, m, when the prepared flexible TENG was implanted into the surface of the femur of SD rats and the legs moved regularly through pulling action, the open-circuit voltage, short-circuit current, and transfer charge generated by the implanted TENG in the body reached about 60 mV, 1 nA, and 0.04 nC, respectively. This self-powered electric stimulator with excellent performance can promote the adhesion, proliferation, and differentiation of osteoblastic progenitor cells, to the point of regulating the calcium ion in them.

In 2018, Yao et al. proposed an implanted vagus nerve stimulation system, which required no battery and had a spontaneous response to gastric movement (Figure 6n) [118]. When the TENG was attached to the gastric wall of rats, with the movement of the gastric wall, the triboelectric layer (PTFE and Au) and electrode layer (AU) would come into contact and separate to produce biphasic electric pulses. Bilateral VNS near the gastroesophageal junction were wrapped by Au leads. The Cu line electrode was connected to the Au cable in order to transmit electrical signals. PDMS and Ecoflex were used as packaging layers with good biocompatibility, mechanical strength, and flexibility. The TENG electrode was directly related to the vagus nerve, where the tension signal generated by the gastric motion stimulated the vagus nerve to reduce dietary consumption. The VNS system quickly achieved 35% weight loss in 18 days and maintained 38% during the remaining 75 days of testing.

Hinchet et al. demonstrated an implanted vibration–friction electricity generator that effectively collected mechanical energy transmitted through the skin and liquid [124]. The high-frequency-vibration implantable triboelectric generator (VI-TEG) was composed of PFA film (approximately 50 μm thick) as a negative friction layer, a 3.6 cm × 3.6 cm thin copper electrode was fabricated on a flexible printed circuit board (PCB) as a positive friction layer, and the air gap was 80 μm. Then, the film was sealed with molten adhesive. The thickness of VI-TEG was less than 1 mm. In pig tissue, the voltage and current of VI-TEG produced by ultrasonic energy transfer in vitro reached 2.4 V and 156 μA. This achievement is the first technology that can compete with piezoelectricity to harvest ultrasound in vivo and power medical implants.

It is well known that chemotherapy for cancer has serious side effects and low efficacy. In order to overcome the shortcomings of drug delivery systems (DDSs) in the past decades, in 2019, Zhao et al. demonstrated the first controlled DDS supported by TENG for cancer treatment and achieved excellent antitumor efficacy in vivo [119]. An encapsulated magnet TENG (MTENG) with a new structure was fabricated to control DOX-loaded red blood cell (D@RBC) drug release. PTFE and titanium were used as two friction layers. A 200 nm-thick Cu film was magnetron-sputtered on the back of the PTFE as an electrode. Two magnets were fixed to the back of the PTFE film and titanium sheet to separate the triboelectric layer. External PTFE and PDMS films were used as encapsulation layers to protect MTENG from harsh environments. The authors established a controllable DDS (Figure 6o). Under the electric stimulation of MTENG, the release of DOX increased significantly and returned to normal following stimulation. MTENG-controlled DDS achieved excellent cancer-cell-killing in vitro and in vivo at a low DOX dose.

Whether monitoring physiological phenomena for therapeutic purposes or purely sensing purposes, the electrical output implanted with TENG can produce different biological effects, such as directional growth of nerve cells, pulsatile rate regulation clusters of cardiomyocytes, medical monitoring, and effective weight control. The rapid development of implantable TENG shows its great potential to be applied in biomedical electronics and healthcare surveillance.

## 4. TENG as Power Sources for Biosensors

TENG is not only used to monitor physiological or environmental signals, but can also promote a self-powered sensor system with significantly miniaturized energy storage components, and is a very promising sustainable energy harvester for wearable, implantable electronics and third-party biomedical sensors.

### 4.1. Heart Rate Monitoring

Heart rate, which directly reflects the health of the human cardiovascular system, is one of the vital detection signals most commonly used in the monitoring and diagnosis of human health care. In recent years, technology based on body sensor networks (BSNs) has greatly promoted personalized health monitoring and evaluation, and disease diagnosis. Therefore, the development of a heart rate BSN platform will be a good thing. Thanks to this platform, the heart rate signal can be picked up, transmitted, and displayed in real-time. Using TENG as the power supply of a wearable heart rate sensor makes the equipment not only have the accuracy of a commercial heart rate sensor, but also have the characteristics of a self-power supply and low cost of TENG.

Lin et al. reported a low-structure TENG−driven wireless BSN for non-invasive real-time monitoring of human heart rhythm [73]. The system consisted of four modules: a power management circuit, commercial heart-rate sensor, signal-processing unit, and Bluetooth for data transmission (Figure 7a–c). When a TENG is equipped on the human body, it can remove the inertial energy generated by the human body’s natural walking as the arm swings naturally, and the collected human biological mechanical energy can drive the BSN independently. To make the BSN work as a real-time heart rate monitor, the heart rate signal is first acquired through a signal-processing circuit, and then transmitted to a smartphone for display through Bluetooth wireless technology. This research not only significantly advanced TENG into a system level for portable medical devices, but also offered a favorable solution for tracking personal heart rate.

Maharjan et al. developed a novel, curvilinear, wearable hybrid electromagnetic triboelectric nanogenerator (WHEM−TENG) operating as a fully enclosed, lightweight, low-frequency energy harvester driven by the mechanical motion of a swinging arm, as shown in Figure 7d [125]. A spherical magnetic sphere inside a hollow curved 3D-printed acrylonitrile butadiene styrene (ABS) tube and two copper coils connected in series make up the electromagnetic generator (EMG), wound around the ends of the tube. This WHEM−TENG was also used to drive a commercial heart-rate sensor for monitoring the live heart-pulse from a finger. As shown in Figure 7e,f, using a 0.1 F capacitor for energy storage, it is possible to make the sensor work continuously for 6 s after 10 min of running activity. Therefore, wearable hybrid electromagnetic triboelectric nanogenerators have bright development prospects in the future.

### 4.2. Body Condition Monitoring

Early in 2017, Dong et al. prepared an all-yarn-based self-powered knitting textile that could collect human motion energy and power a temperature–humidity meter [71]. The three-ply twisted stainless-steel/polyester fiber-blended yarn was chosen as the electrode, and silicone rubber acted as a dielectric layer. This TENG fabric could light up 124 light-emitting diodes when manually tapped; moreover, by adding a full-wave rectifier, the device could drive a temperature–humidity meter from continuous hand tapping. Guo et al. proposed a cut-paper architecture TENG that can simultaneously harvest and storing energy [126]. By combining this TENG and a supercapacitor with a rectifier, the self-charging power unit could be easily placed in a wallet. Furthermore, just by hand-flapping motion, the self-powered system could drive a commercial temperature sensor (Figure 7g,h) As shown, a temperature sensor could be charged to ~1.45 V in 63 s and maintain stable at a working voltage of ~1.35 V. These works demonstrate the potential of TENG as a power source for portable sensors.

In 2019, He et al. developed a wearable textile nano-energy nano-system (NENS) for medical applications [127]. They skillfully integrated TENG with a high-voltage diode and mechanical switch to adjust the charge flow in the system so as to improve the triboelectric output of the flexible TENG. The nano-device consists of two layers of conductive textiles covered with triboelectric materials. A thin, wrinkled nitrile film was one of the triboelectric materials, and another was silicone rubber. The working mechanism of the diode-amplified T−TENG (D−T−TENG) is shown in Figure 7i; the high-voltage diode connected in parallel with the TENG could effectively accumulate charge at both ends of the diode, resulting in high open-circuit voltage. The results show that D−T−TENG can produce a high closed-loop current 25 times higher than that of the T−TENG alone, and the charging speed of the capacitor also increased by about four times. Furthermore, the enhanced output current could easily achieve controlled muscle stimulation. In addition, D−T−TENG can also harvest the energy in normal walking to power the Bluetooth module for ambient humidity and temperature sensing (Figure 7j–m).

Song et al. designed a TENG−powered wearable sweat sensor system compatible with traditional flexible printed circuit board (FPCB)-manufacturing processes that can dynamically monitor key sweat biomarkers (e.g., pH and Na^+^) (Figure 7n) [128]. The TENG consisted of an interdigital stator and a grating-patterned slider. PTFE and copper were used as tribo-pairs. With an electroless nickel/immersion gold (ENIG) surface finish on the electrode area, the stator was further laminated by PTFE. Then, the sweat sensor patch was designed based on ion-selective electrodes. For pH analysis, the deprotonation of H^+^ atoms on the surface of the electrodeposited PANI layer was an indicator of the bulk H^+^ concentration. Na^+^ concentration measurements were facilitated by a thin ion-selective membrane containing a Na^+^ ionophore X, and a poly-(3,4-ethylenedioxythiophene (PEDOT):poly-(sodium 4-styrenesulfonate) (PSS) layer in between the gold electrode layer and sodium ion-selective membrane as an ion-electron transducer. Through seamless system integration and efficient power management, this system could power multiplexed sweat biosensors and wirelessly send data to the user interfaces through Bluetooth during on-body human trials. This technology could serve as an attractive approach toward self-powered wireless personalized health monitoring during people’s daily activities with further development.

In 2022, Xu et al. manufactured a double−TENG−actuated moisture sensor for ambient temperature moisture detection [129]. This dual−TENG consisted of a backing plate, a sponge, a double-layer conductive copper sheet, and an EFF film. The peak-to-peak voltage of TENG may have been near 700 V, and the maximum output power was 12.208 mW. A flexible chitosan/activated carbon composite sensor was integrated with the dual−TENG to form a self-fueled moisture-detection system. The sensor had a large sensing range (0~97% relative humidity) and good stability. Its response/recovery time under 33% RH and 97% RH conditions were 12 s/20 s and 22 s/21 s, respectively. As we know, human respiration can cause important changes in humidity around the nose and mouth. These changing moisture values often contain information on an individual’s fitness. It is, therefore, quite convenient to control these signals.

### 4.3. Drug Delivery

Transdermal drug delivery (TDD) systems with feedback control have generated a great deal of research and clinical interest due to their unique convenience, self-management, and safety. In particular, advances in wearable biosensors enable ongoing, non-invasive, real-time surveillance of human activities, as well as biomarkers that can prevent pain and irritation. The biosensing signals collected (e.g., glucose concentration and strain induced by human motion) may be used as feedback control of TDD to obtain treatment upon request. Therefore, the integration of portable biosensors and TDD provides solutions for precision medicine by carrying out the surveillance and treatment of closed-loop diseases.

It should be noted that antibacterial and biocompatible qualities must be taken into account when using TENG for the administration of medicinal products in live animals in order to avoid any potential adverse effects, such as inflammation. In 2018, Zhang et al. manufactured a “genetically modified” bio-functional TENG using recombinant spider silk proteins (RSSP) with extraordinary antibacterial performance in vitro and in vivo [130]. The self-powered RSSP patch was built by casting the RSSP solution, which was functionalized by graphene, carbon nanotubes, and drug molecules, on a PET/ITO substrate and then assembling it with another PET/ITO substrate to form a self-powered TENG patch. The antibacterial mechanism of the RSSP patch is demonstrated in Figure 8a. The triboelectrically charged RSSP patch constructs a potential difference between the bacteria and the positively charged surface. Subsequently, the transmission of extracellular electrons between the bacteria and the RSSP patch affects the morphology of the bacteria, contributing to the death of the bacteria after loading. The authors also studied the biocompatibility of the RSSP patch in vivo by placing it on a wound of mice infected with staphylococcal aureus. Following suture and culturing for 7 days, a high antibacterial level of 67.4% was shown, indicating the effective antibacterial ability of the RSSP patch. This work provides a new TENG based on high-performance biomaterials and expands its potential for multipurpose applications.

Wu et al. developed a self-powered TDD iontophoretic system that can be controlled and regulated by the energy obtained from biomechanical movements [131]. This system was composed of a portable TENG as an integrated power source and a fix based on hydrogel as a drug carrier. As shown in Figure 8b,c, the insole TENG used PTFE as one triboelectric material, Al as the other triboelectric material and electrodes, polyethylene terephthalate (PET) as the substrate, and Kapton as the spacer. Using a unique representational unit with the drug patch on the ankle and a TENG insole under the shoe, the medication can be given upon request when the patient is walking on their foot and producing electricity at the same time. Moreover, when using Rhodamine 6G dye as the model drug loaded in one hydrogel cell, using pig skin to surrogate human skin, the drug patch was connected to the TENG via a rectifier, with the electrode on the dye-loading cell wired to the positive terminal and the other electrode to the negative terminal. About 320 nC of load was transferred per operating cycle, and a peak-to-peak AC output voltage with 8 V amplitude was reached. Following the rectifier, pulsed DC current with a maximum of 12 µA was delivered to the engineered hydrogeological device at a maximum voltage of 4 V. During operation, the TENG electrical outlets would drive the ions inside the hydrogel drug plate to circulate between cells. These studies favor autonomous TENG−based systems toward advanced biomedical treatment.

Ouyang et al. introduced a TENG−based portable TDD system for the accurate administration of medicines and upon request [132]. The system was made up of three components: transdermal patches (pharmaceutical patch and ionophoresis patch), TENG electrodes, and the power-management circuit. Inside the drug patch, a screen-printed electrode (SPE) coated with a thin film of the conductive PPy nanocomposite was used as a medication reservoir (Figure 8d–f). The TENG was designed as a rotary type, and manually rotating the TENG (30–40 rpm) for 1.5 min could release a drug dosage of 3 μg cm^−2^. Moreover, the speed could be set from 0.05 to 0.25 μg cm^−2^ per minute by changing the TENG load duration or the resistance of the power-management circuit. Additionally, ex vivo swine skin experiments validated the performance of this TENG−based drug delivery system with a 50% improvement over conventional transdermal patches.

TRGs have been the subject of extensive studies as energy-capture devices, but their use in powering biomedical sensors requires further study. For example, the durability of the proposed TENG−powered biosensing devices must be examined in depth [70]. In addition, some ubiquitous commercial sensors have very high energy consumption, which can hinder their marketing. Ultimately, however, they have the potential to integrate a variety of communication technologies and networks, such as IoTs, embedded chips, and artificial intelligence, as well as technology, such as data fusion, data exploration, and parallel, and distributed algorithms, for the benefit of mankind.

## 5. Conclusions and Perspectives

Overall, we summarize current developments in the area of TENG polymer-based biosensing systems, while demonstrating the broad potential for future medical applications. The biosensor based on TENG not only shows an improved wearable shape, which can be directly attached to the skin, passively integrated with textiles, or actively interwoven in textiles, but can also be operated in vivo and in vitro. TENG’s electrical output contains rich physiological and biomedical information, which allows TENG itself to be used as a self-powered biosensor. More importantly, TENG can also be used as a power supply for sustainably driving biomedical sensors. In the current situation, although TENG−based biosensors have limitations in integrating and commercializing, there is great potential for making universal and personalized medicine possible in the IoT era.

### 5.1. Power Output Enhancement

At present, there are three main strategies to improve the power output of devices, including surface modification, material selection, and device structure. These methods can effectively improve the energy conversion of these devices.

Currently, changing the surface of the triboelectric layer involves primarily physical, chemical, biological, and other methods to improve the capability of TENG. Methods of physical alteration include the modeling method, plasma processing, and engraving. Chemical modification is primarily the grafting of certain chemical groups that produce electrons or accept electrons. While biomaterials and biotechnologies are primarily used for the biomodification of TENGs, the use of biomodification can make TENG biocompatible and biodegradable, which is critical for biomedical screening devices and can largely solve the problem of the effects of the immune system response on foreign bodies. In total, the three above change methods may be combined to further improve efficiency. At the same time, new methods of alteration also require further exploration and research.

The properties of the materials themselves are critical for high-performance TENG−based biosensor applications. In addition to the commonly used negative triboelectric materials, such as PDMS, PVDF, and PTFE, efforts should be made to develop new triboelectric materials, which should have high performance, stability, biocompatibility, and versatility.

The properties of the materials themselves are critical for high-performance TENG−based biosensor applications. Most of the materials used are polymer materials. As known, these polymers possess various functional groups, such as fluorine (-F), carboxyl (-COOH), amidogen (-NH_2_), cyano groups (-CN), ester groups (-COOR), alkoxy (-OR), hydroxyl (-OH), acyl groups (-CON-), nitro (-NO_2_), and amide groups (-CONH). The presence of these functional groups facilitates electron transfer and electron capture in the triboelectrification process. Now, PDMS, PVDF, FEP, PTFE, and so on are commonly used negative triboelectric materials because of the strong electron attractive force of the fluorine element. Polymers containing electron-donating groups, such as PVA, Ppy, TPU, etc., are always used as positive triboelectric materials. However, facing more and more different application scenarios, such as in liquid, in vivo, and even in high temperature and high-pressure conditions, efforts should be made to develop new high-performance polymer triboelectric materials.

The structural design of the TENG also affects the responsiveness of the sensor, and the size of the TENG is further optimized to match that of the human body tissue. Future research can focus on the optimization of the structure of the device and the improvement of the anti-interference capability.

### 5.2. Durability and Stability

Stable output is one of the main parameters for the realistic application of TENG in real-world settings. Since TENG generates electrostatic charges based on the friction and contact between layers during the working process, bending and stretching in long-term applications will cause wear and erosion to the material. Therefore, future research directions can focus on testing new materials that are even more wear-resistant. In addition, future research can be devoted to developing new materials with high friction resistance. For instance, carbon materials are promising candidates due to their excellent properties. Additionally, the self-healing capabilities of devices, especially in wearables, may be a major concern. Additionally, surface modification through the material may be the decisive parameter to enhance the output manifestation of TENG in the long-term. In addition, more attention could be devoted to 3D printing designs to simplify the development process and to developing finely fabricated TENG arrays to strengthen connections between arrays, leading to the development of more flexible materials. Therefore, more research should be carried out on optimizing material properties.

Moreover, the impact of high temperature, high humidity, or a closed human environment on the equipment needs to be considered, which requires more attention to be paid to the sustainability of materials during use. Therefore, the function of daily packaging can be realized by adding waterproof and washable materials. Therefore, future research can focus on improving sealed packaging and integrating different hydrophobic membranes on TENG to increase protection through the hydration layer. Another approach is to develop self-healing medical sensing devices consisting of repairable materials. Self-repairing materials are sought after because of their inherent resistance to mechanical bending, distortion, extrusion, and sliding, which makes them promising candidates for long-term wearable applications. In addition, the anti-noise characteristics of wearable TENG are also important due to a large amount of interference in human movement, because this movement will inadvertently produce interference signals, which will reduce the sensing performance. In fact, in order to further improve the noise immunity, future research should focus on the circuit design combining low-pass and high-pass filtering, and so on.

### 5.3. Multifunctionality

An important challenge mentioned above is the versatility of implantable TENG (I−TENG) to realize detection, drug supply, and treatment at the same time. In order to match the TENG with the tissue at the implantation site, the size of the TENG needs to be further reduced. Furthermore, the position of the implantable device in the target tissue must be capable of producing sufficient movement for adequate performance. For this problem, this is an important challenge for I−TENG, because it mainly uses the contact separation mode. In this context, additional research should be envisaged in the future to address this key issue in large animal models. As a feasible scheme, first of all, miniaturize the equipment and make it an integrated system to realize incursive surgery so as to reduce the risk of infection and hurt to patients, and reduce pain, which is also a promising option. In future research, we can focus on the development of a miniaturized cluster TENG, which can not only increase the surface area contained in the TENG array, but also reduce the overall immune response. Reducing invasion while maximizing power generation can promote the development of this field.

### 5.4. Comfortability

Wearable medical devices should hardly affect users, or even disappear in the artificial node devices around us, so that people will accept and be willing to use wearable medical devices to realize universal and personalized medicine. In view of this, it is necessary to make TENGs lightweight, breathable, miniaturized, and biocompatible. However, the output performance cannot be affected at the same time. Metal coatings and wires are currently used in textile TENGs, which may affect comfort. Potentially popular solutions are the use of advanced materials, such as silk and cotton, to ameliorate breathability and contribute a lightweight solving method, and use more biocompatible materials to guard against even more damage and complications after use. While modifying the material and device structure can effectively meet comfort expectations, the electrical conductivity of the material still needs to be further optimized.

Despite these challenges, TENGs are developed with advanced properties, such as self-powering, low cost, biocompatibility, very high sensitivity, and output responsive to mild stimuli. With the current trend of developing electronic devices into portable forms, the perspective of TENG−based biosensing systems should shine on the health market. It is hoped that this review will provide insight into current TENGs as sources of inspiration for future biomedical research to improve human life.

## Figures and Tables

**Figure 1 biosensors-12-00323-f001:**
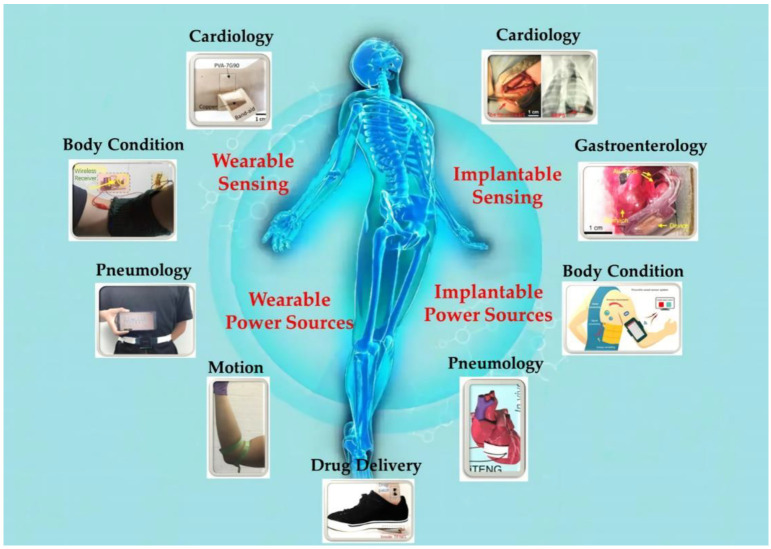
Schematic illustration of TENGs for biosensing, which includes wearable, implantable, and TENG as an external power source for third-party biosensors.

**Figure 2 biosensors-12-00323-f002:**
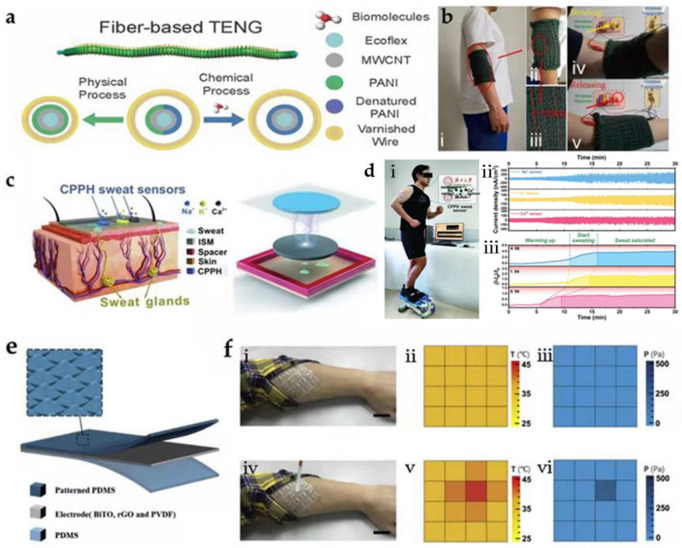
Applications of wearable TENG−based biosensors for body condition monitoring. (**a**) The structure of a single TENG. (**b**) Smart clothing integrated with TENG (i–v). Insets iv and v show a wireless control system integrated with TENG. Reprinted/adapted with permission from Ref. [75]. 2022, Zhao et al. Elsevier. (**c**) CPPH sweat sensor placed on cutaneous sweat glands to detect and quantify Na^+^, K^+^, and Ca^2+^ and the schematic of the structure of a CPPH sweat-sensor module. (**d**) Real-time sweat-ion monitoring on the subject during exercise on a mini stair stepper (i–iii). Insets ii and iii show the current density curve and current ratio curve of the CPPH sweat sensor of Na^+^, K^+^, and Ca^2+^ within 30 min of exercise of the subject. Reprinted/adapted with permission from Ref. [76]. 2022, Qin et al. Wiley-VCH. (**e**) Schematic illustration showing the basic structure of the tactile e-skin. (**f**) (i) Optical image showing the e-skin mounted on a person’s arm, (ii) recorded distribution map of the temperature, and (iii) recorded distribution map of the pressure. (iv) Optical image showing an electric heater touching the e-skin mounted on the arm, (v) recorded distribution map of the temperature, and (vi) recorded distribution map of the pressure. Reprinted/adapted with permission from Ref. [77]. 2020, Rao et al. Elsevier.

**Figure 3 biosensors-12-00323-f003:**
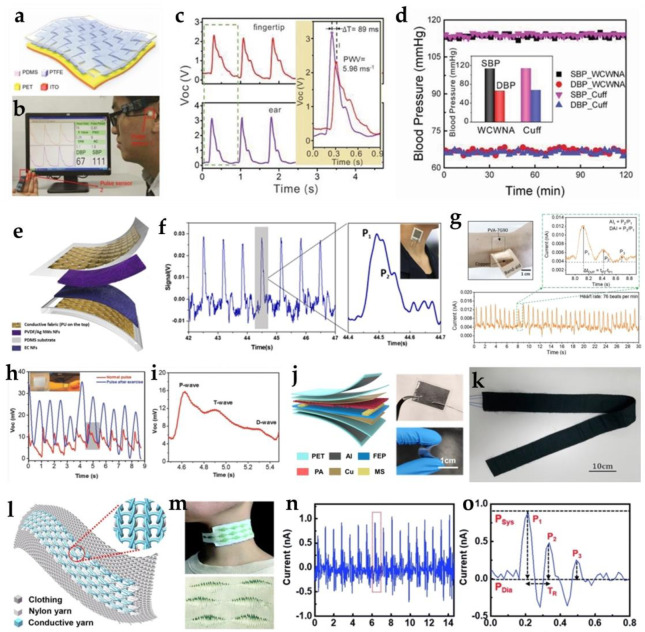
Applications of wearable TENG−based biosensors for cardiology. (**a**) Schematic illustration of the flexible weaving-constructed self-powered pressure sensor. (**b**) Photograph showing the sensor system with two pulse sensors on the human fingertip and ear for PWV and BP detection. (**c**) Corresponding voltage signals from the two pulse sensors. The inset shows an enlarged view of the voltage signal in one pulse cycle. (**d**) Systolic blood pressure (SBP) and diastolic blood pressure (DBP) measurements. Reprinted/adapted with permission from Ref. [78]. 2019, Meng et al. WILEY-VCH. (**e**) Structural design of the pressure sensor textile. (**f**) Real-time detection of the pulse wave of the carotid artery using the sensor textile. (magnified pulse wave consisting of two peaks corresponding to the pulse pressure (P1), which is the difference between the systolic (P_sys_) and diastolic (P_dia_) pressure, and the reflected wave pressure produced from the hand (P2). Reprinted/adapted with permission from Ref. [79]. 2020, Lou et al. American Chemical Society. (**g**) Optical image of the PVA film attached to the human wrist (left). Thirty-second real-time current outputs from the PVA-based TENG device measuring the pulses (right). One pulse period was enlarged (bottom). Reprinted/adapted with permission from Ref. [80]. 2020, Wang et al. WILEY-VCH. (**h**) Real-time arterial pulse waves under normal and exercise conditions. The inset is a photograph of a pressure sensor attached to a wrist. (**i**) Single-signal waveform extracted from the marked region in (**h**). Reprinted/adapted with permission from Ref. [26]. 2018, Dong et al. WILEY-VCH. (**j**) Schematic structural diagram of the SUPS; insets are optical photographs of a constructed SUPS. Reprinted/adapted with permission from Ref. [81]. 2021, Xu et al. Elsevier. (**k**) Tactile sensor made into a sleep-monitoring belt. Reprinted/adapted with permission from Ref. [82]. 2018, Ding et al. MDPI. (**l**) Schematic illustration of the combination of TATSA and clothes. The inset shows an enlarged view of the sensor. Reprinted/adapted with permission from Ref. [4]. 2020, Fan et al. Science. (**m**) Photographic images of the fashionable garment-type triboelectric sensor with arbitrary stitch patterns for pulse pressure detection. (**n**,**o**) Variation in the pulse pressure wave before and after physical exercise. Reprinted/adapted with permission from Ref. [85]. 2020, Shin et al. The Royal Society of Chemistry.

**Figure 5 biosensors-12-00323-f005:**
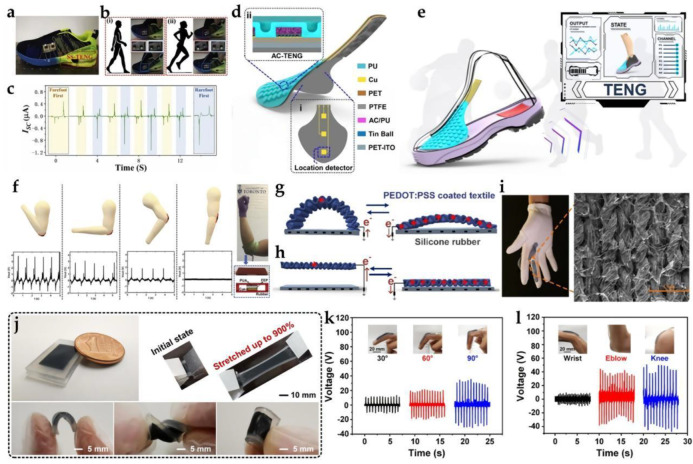
Applications of wearable TENG−based biosensors for motion. (**a**) Photograph of the SS−TENG installed into a common shoe; (**b**) Demonstration of a self-powered motion sensor by integrating LED circuit boards; (**c**) Current signals of SS−TENG under forefoot strike and rare foot strike. Reprinted/adapted with permission from Ref. [99]. 2019, Lin et al. Elsevier. (**d**) Two functional units constituting ISSP and their corresponding TENG of different modes: (i) layout diagram of location detectors on ISSP; (ii) internal structure diagram of an air-capsule TENG. (**e**) Dynamic detection applications based on IoT sensory system. Reprinted/adapted with permission from Ref. [97]. 2022, Yang et al. American Chemical of Society. (**f**) Effect of arm bending angle on the output signal of the PUA−TENG sensor. The right-hand-side inset shows the schematic and fabricated device. Reprinted/adapted with permission from Ref. [100]. 2019, Saadatnia et al. Elsevier. (**g**) Working mechanism of the arch-shaped TENG under the stretching and releasing state. (**h**) Working mechanism of the contact–separation TENG sensor. (**i**) Photograph of the glove-based interface. The inset shows an SEM image of the PEDOT: PSS-coated textile. Reprinted/adapted with permission from Ref. [101]. 2019, He et al. Elsevier. (**j**) Mechanical characterization of the super-stretchable TENG. (**k**) Proposed SS−TENG employed to detect the bending angles of knuckle. (**l**) SS−TENG used to qualitatively distinguish the motions of different human joints, including the wrist, elbow, and knee, according to the differences in amplitude and waveform of the output voltage. Reprinted/adapted with permission from Ref. [102]. 2021, et al. Elsevier.

**Figure 6 biosensors-12-00323-f006:**
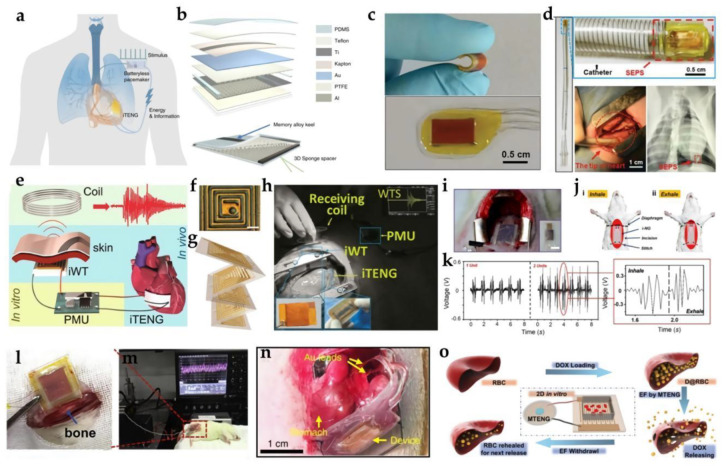
Implantable TENGs for cardiological biomonitoring. (**a**) Illustration of symbiotic cardiac pacemaker system. (**b**) Schematic structural diagram of iTENG. Reprinted/adapted with permission from Ref. [114]. 2019, Ouyang et al. Nature. (**c**) Photograph of the SEPS in the bending and original states. (**d**) Photograph of minimally invasive surgery with a digital radiography image of the heart implanted with a device by integration with a surgical delivery system. Reprinted/adapted with permission from Ref. [115]. 2019, Liu et al. WILEY-VCH. (**e**) Schematic diagram of the self-powered wireless transmission system based on the iTENG (iWT, implantable wireless transmitter; PMU, power management unit; WTS, wireless transmission signal). (**f**,**g**) Photograph of the implantable wireless transmitter. (**h**) In vivo heart rate monitoring (inset: enlarged view of the implantable wireless transmitter, iTENG, and wireless transmission signal). Reprinted/adapted with permission from Ref. [116]. 2016, Zheng et al. American Chemical of Society. (**i**) Digital image of an i−TENG implanted inside the abdominal cavity of a SD rat. The inset is an image of the i−TENG device. (**j**) Working process of the i−TENG driven by diaphragm motion during respiration. (**k**) In vivo voltage outputs measured from i−TENGs with a single unit (left) and double unit (right). The right panel is an enlarged voltage output profile within one inhale–exhale cycle. Reprinted/adapted with permission from Ref. [117]. 2018, Li et al. American Chemical of Society. (**l**) Surface of femur region implanted with the flexible TENG. (**m**) Image of the output measurement process; a linear motor was used to pull the leg of an SD rat through a line [19]. (**n**) Implanted VNS device with Au leads being connected to anterior and posterior vagal trunks. Reprinted/adapted with permission from Ref. [118]. 2018, Yao et al. Nature. (**o**) Schematic illustration showing the loading of DOX into RBCs and the subsequent integration of MTENG to realize controlled DOX release. Reprinted/adapted with permission from Ref. [119]. 2019, Zhao et al. WILEY-VCH.

**Figure 7 biosensors-12-00323-f007:**
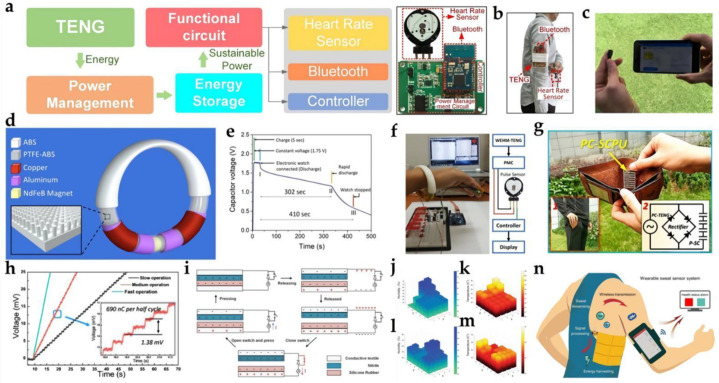
TENGs as power sources to drive biosensing devices. (**a**) System diagram of the complete power-supplying system. (**b**) Photograph showing that the integrated system was equipped onto a human arm. (**c**) Photograph showing that the acquired heart-rate data were wirelessly transmitted to a cell phone in real-time. Reprinted/adapted with permission from Ref. [73]. 2017, Lin et al. American Chemical of Society. (**d**) Schematic diagram of the fabricated hybrid nanogenerator. Inset: sketch of the nanostructured polytetrafluoroethylene film at an angle of 30°. (**e**) Voltage charging and discharging curves for a 1000 µF capacitor during driving a wrist-watch, showing starting phase I, rapid discharging phase II, and watch stop phase III. (**f**) Demonstration of heart-rate and pulse-signal monitoring powered by the hybrid nanogenerator; inset showing a diagram of a heart-rate-monitoring system. Reprinted/adapted with permission from Ref. [125]. 2018, Maharjan et al. Elsevier. (**g**) Photograph of the practical application of a PC-SCPU placed in a wallet. Inset 1 shows that an ultralight PC-SCPU can be easily placed in the pocket. Inset 2 shows the electric circuit loop of the PC-SCPU. (**h**) V–t curve of the PC-SCPU with charging by hand flapping (from slow operation to fast operation). Reprinted/adapted with permission from Ref. [126]. 2017, Guo et al. American Chemical of Society. (**i**) Working mechanism of the D−T−TENG. (**j**,**k**) Humidity and temperature distribution of the lab environment with 20 sampling points. (**l**,**m**) Humidity and temperature distribution of the apartment with 24 sampling points. Reprinted/adapted with permission from Ref. [127]. 2019, He et al. Wiley Online Library. (**n**) Schematic illustrating the FWS^3^ that integrates human motion energy harvesting, signal processing, microfluidic-based sweat biosensing, and Bluetooth-based wireless data transmission to a mobile user interface for real-time health-status tracking. Reprinted/adapted with permission from Ref. [128]. 2020, Song, Science.

**Figure 8 biosensors-12-00323-f008:**
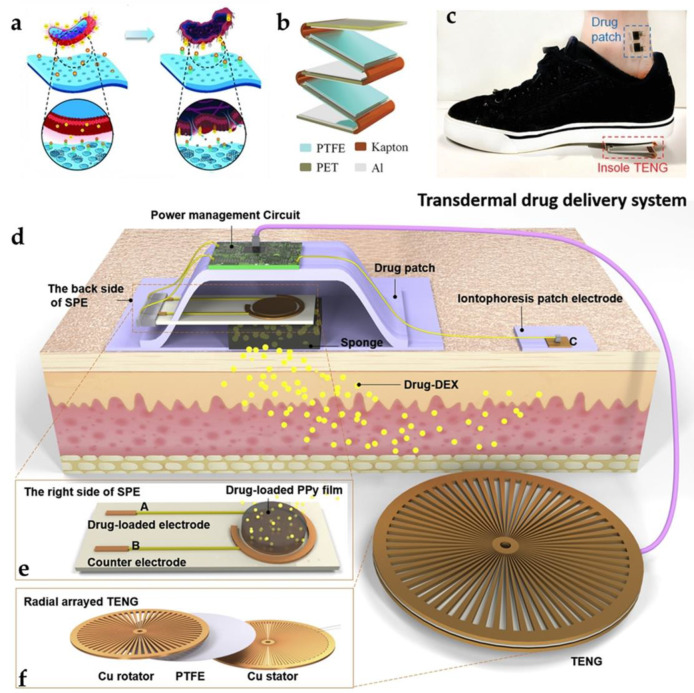
TENGs as power sources for drug delivery. (**a**) Diagram showing the antibacterial mechanism. Reprinted/adapted with permission from Ref. [130]. 2018, Zhang et al. Wiley Online Library. (**b**) Schematic of the insole TENG used in this study. (**c**) Photograph demonstrating the idea of a wearable TDD system on a human ankle consisting of the insole TENG, and the designed drug patch. Reprinted/adapted with permission from Ref. [131]. 2019, Wu et al. Wiley Online Library. (**d**) System consisting of transdermal patches (drug patch and iontophoresis patch electrode), TENG, and power-management circuit. (**e**) Right side of the SPE. (**f**) Radial-arrayed rotary TENG consisting of two copper layers and one PTFE layer. Reprinted/adapted with permission from Ref. [132]. 2019, Ouyang et al. Elsevier.

## Data Availability

Data available on request from the authors.

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
