# Peer review of "From Triboelectric Nanogenerator to Polymer-Based Biosensor: A Review"

_biosensors, 2022, doi:10.3390/bios12050323_

Round 1

Reviewer 1 Report

This manuscript presents a deep and extensive review of the research and developments that have emerged during approximately the last 5 years with respect to TENGs based self-powered biosensors. All topics considered are appropriate, timely and well targeted. The number and quality of the references are adequate and reflect well the review carried out. The language is mostly precise and correct. I recommend an acceptance after minor revision because there are some writing issues in the manuscript.

  1. There are some missing spaces through the text. For example, “0.35μA and 45.8μW cm–2” in Line 343.
  2. To express “per”, some units use superscript, but some units use slash. This should be united
  3. Line 329, the abbreviation “TPU” is not described.
  4. “Fish gelatin” firstly appears in Line 339, but the abbreviation “FC” is defined in Line 340.
  5. Line 332, “With this TENG, a 47 μF commercial capacitor can be charged 4.5 V”, is the preposition “to” lost between charge and 4.5 V?
  6. Line 335, in the sentence “what’s more, an alarm would be triggered when when the tester’respiratory signal was below threshold”, the word “when” is written twice, and the sentence is confusing.
  7. In some sentences, the comma is lost in front of the word “and”.

Author Response

Comment 1:

1. There are some missing spaces through the text. For example, “0.35μA and 45.8μW cm-2” in Line 343.

Response: We express great thanks for your professional review work on our manuscript. We have corrected our improper statement according to your good suggestion. Thank you very much.

2. To express “per”, some units use superscript, but some units use slash. This should be united

Response: We really appreciate your valuable comments. In accordance with your good suggestions, we revised the inappropriate unit expression in the revised draft to keep the unit of the full text unified. thank you very much.

3. Line 329, the abbreviation “TPU” is not described.

Response: Thank for your suggestions. We really appreciate your valuable comments, which are of great help to our work. According to your instructions, we supplemented the description of "TPU" in the original. Thank you.

4. “Fish gelatin” firstly appears in Line 339, but the abbreviation “FC” is defined in Line 340.

Response: According to the reviewer's good instruction, we have carefully revised the words. Thanks for your good comments.

5. Line 332, “With this TENG, a 47 μF commercial capacitor can be charged 4.5 V”, is the preposition “to” lost between charge and 4.5 V?

Response: According to your good instructions, we have modified the content of all of the above. Additionally, we have examined the whole manuscript carefully and tried to avoid similar mistakes. Thank you very much.

6. Line 335, in the sentence “what’s more, an alarm would be triggered when when the tester’respiratory signal was below threshold”, the word “when” is written twice, and the sentence is confusing.

Response: We really appreciate your valuable comments. In accordance with your good suggestions, we have modified improper words in revised manuscript. Thank you very much.

7. In some sentences, the comma is lost in front of the word “and”.

Response: According to your good instruction, we have added the corresponding comma in our revised manuscript. Special thanks for your good comments.

Reviewer 2 Report

Comment for Biosensors Journal (ID: biosensors-1713838)

This work reviewed the self-powered wearable biosensors that are based on triboelectric nanogenerators (TENGs) for healthcare. However, this reviewed manuscript has faced some main issues that should be completely revised followed by the Minor Revision to be more obvious and concise in understanding addressing the following issues:

  1. Please maintain the content regularity in this manuscript. For instance, the word size in line 20-21is larger than in other lines.
  2. This study has been focused on polymer materials and their functionalities. Therefore, it would be better if you roughly illustrate the positive effect of utilizing the polymer-based material in TENG.
  3. Some figures in this review are disordered such as Figure 3 which has included the application of the TENG-biosensor. The structure and functions of sensor in the individual cited study should be fit together instead of randomly placing the figures which has similar functions.
  4. Attempt to reorganize the content for the sake of clear and concise understanding. The main topics in this review are almost similar and ambiguous such as TENG as power sources for biosensor. The core issue is the self-powered sensor utilizing power from TENG for health monitoring. Please explain why it should be necessarily separated as one of the main topics.

Author Response

Comment 2:

1. Please maintain the content regularity in this manuscript. For instance, the word size in line 20-21is larger than in other lines.

Response: Thanks for you careful checks. We are sorry for our carelessness. Base on your comments, we have made the corrections. Thank you very much.

2. This study has been focused on polymer materials and their functionalities. Therefore, it would be better if you roughly illustrate the positive effect of utilizing the polymer-based material in TENG.

Response: We very appreciate your valuable comments. In accordance with your good suggestions, we have added some discussions in revised manuscript. Thank you very much.

3. Some figures in this review are disordered such as Figure 3 which has included the application of the TENG-biosensor. The structure and functions of sensor in the individual cited study should be fit together instead of randomly placing the figures which has similar functions.

Response: Thanks for your good suggestions. According to your good instructions, we have made modifications of figures in revised manuscript. Thank you very much.

4. Attempt to reorganize the content for the sake of clear and concise understanding. The main topics in this review are almost similar and ambiguous such as TENG as power sources for biosensor. The core issue is the self-powered sensor utilizing power from TENG for health monitoring. Please explain why it should be necessarily separated as one of the main topics.

Response: We indeed appreciate your valuable comments.  Over the past 30 years, biomedical sensors have played a vital role in improving the effectiveness of health care, although limited power access and user wearability restrictions prevent them from playing a sustained and active biomedical sensing role in our daily life. Triboelectric nanogenerator (TENG) shows extraordinary ability and versatility in providing self powered and wear optimized biomedical sensors, and paves the way for a new platform technology that can be fully integrated into the developing 5G/Internet of things ecosystem. This new paradigm of biomedical sensors based on TENG is eager to provide ubiquitous and ubiquitous real-time biomedical sensors for all of us. In this review, we introduce the remarkable development of biomedical sensing based on polymer based TENG in recent ten years, focusing on in vivo and in vivo biomedical sensing solutions. We first introduce TENG as the most relevant biomedical sensor in the field of pneumology and cardiology related to mortality, as well as other biomedical sensing capabilities related to organs, including walking. We also outline implantable biomedical sensing as a growing area of interest in health monitoring. Finally, we discussed the power supply of TENG as a third-party biomedical sensor in many fields, summarized our review by focusing on the future prospects of TENG biomedical sensors, and emphasized the key areas of concern for completely transforming TENG biomedical sensors into clinically and commercially feasible digital and wireless consumer and health products. In addition to their role in monitoring physiological or motion signals, the use of TENG as an energy collection device has further promoted their integration with biomedical sensors. Considering that triboelectric effect generally affects daily materials, TENG can be easily used as energy to meet the growing energy demand of wearable or implantable electronic devices. Since TENG can convert the biomechanical energy of human body, TENG has also been studied as the power supply of third-party biomedical sensors. Therefore, in the third part, we specially summarize the research of TENG as the power supply of biosensors in recent years. Thank you very much.

Reviewer 3 Report

Authors well summarize the self-powered wearable biosensors based on TENG for healthcare applications. First of all, the previous references are fine. English grammar looks fine. Organization for each different pieces are well presented in entire manuscript for review article. Especially, good pictures for each section looks very good to cover previous research work. It is hard to find some errors in entire manuscript. In the conclusion section, there are three perspective so it is good to see the stratergies. Therefore, the manuscript could be recommended to be acceptable with some minor comments.

  1. Figure 1 fonts are not clearly to be seen.
  2. Please put , before "and even self-power" in Line 89. 
  3. There is empty and unnecessary space in Line 325.
  4. No funding information.
  5. Authors might comment the technical limits for PDMS, PVDF, and PTFE materials for this application after Line 1067 if authors know the information.
  6. Please check the font size in Line 21.
  7. Please change keywords for small letters.
  8. Please remove underbar of the e-mail address in Line 11.
  9. Please add which or that before are biocompatible in Line 635.
  10. Please change Figure 5k, l to Figures 5k, l in Line 622.

Author Response

Comment 3:

1. Figure 1 fonts are not clearly to be seen.

Response: Thanks for your good suggestions. According to your good instructions, we have made modifications on figures in revised manuscript. Thank you very much.

2. Please put , before "and even self-power" in Line 89.

Response: We have corrected it according to your suggestion. Thank you very much.

3. There is empty and unnecessary space in Line 325.

Response: According to your suggestions, we have corrected it. Thank you very much.

4. No funding information.

Response: According to your suggestions, we have supplemented the funding information in the original. thank you very much.

5. Authors might comment the technical limits for PDMS, PVDF, and PTFE materials for this application after Line 1067 if authors know the information.

Response: We indeed appreciate your valuable comments, which are very helpful for improving our work. According to your suggestion, we have made more in-depth comments on the technical limitations of PDMS, PVDF and PTFE materials in biosensor applications. Thank you very much.

6. Please check the font size in Line 21.

Response: We are sorry for our carelessness. According to your suggestion, we have corrected it. Thank you very much.

7. Please change keywords for small letters.

Response: Thanks for you help. We feel really sorry for our carelessness.

8. Please remove underbar of the e-mail address in Line 11.

Response: According to the reviewer's good instruction, we have removed underbar of the e-mail address in Line 11. Thank you very much.

9. Please add which or that before are biocompatible in Line 635.

Response: We really appreciate your valuable comments, which are very helpful for improving our work. According to your good instruction, we have added the corresponding content. Thank you very much.

10. Please change Figure 5k, l to Figures 5k, l in Line 622.

Response: We express great thanks for your professional review work on our manuscript. We have corrected our improper statement according to your good suggestion. Thank you very much.
